# The Canonical Wnt Pathway as a Key Regulator in Liver Development, Differentiation and Homeostatic Renewal

**DOI:** 10.3390/genes11101163

**Published:** 2020-09-30

**Authors:** Sebastian L. Wild, Aya Elghajiji, Carmen Grimaldos Rodriguez, Stephen D. Weston, Zoë D. Burke, David Tosh

**Affiliations:** Department of Biology and Biochemistry, University of Bath, Bath BA2 7AY, UK; ae364@bath.ac.uk (A.E.); cgr28@bath.ac.uk (C.G.R.); sdw36@bath.ac.uk (S.D.W.); bsszdb@bath.ac.uk (Z.D.B.); bssdt@bath.ac.uk (D.T.)

**Keywords:** liver, Wnt signalling, zonation, development, differentiation, endoderm, stem cell

## Abstract

The canonical Wnt (Wnt/β-catenin) signalling pathway is highly conserved and plays a critical role in regulating cellular processes both during development and in adult tissue homeostasis. The Wnt/β-catenin signalling pathway is vital for correct body patterning and is involved in fate specification of the gut tube, the primitive precursor of liver. In adults, the Wnt/β-catenin pathway is increasingly recognised as an important regulator of metabolic zonation, homeostatic renewal and regeneration in response to injury throughout the liver. Herein, we review recent developments relating to the key role of the pathway in the patterning and fate specification of the liver, in the directed differentiation of pluripotent stem cells into hepatocytes and in governing proliferation and zonation in the adult liver. We pay particular attention to recent contributions to the controversy surrounding homeostatic renewal and proliferation in response to injury. Furthermore, we discuss how crosstalk between the Wnt/β-catenin and Hedgehog (Hh) and hypoxia inducible factor (HIF) pathways works to maintain liver homeostasis. Advancing our understanding of this pathway will benefit our ability to model disease, screen drugs and generate tissue and organ replacements for regenerative medicine.

## 1. Introduction

The canonical Wnt (Wnt/β-catenin) signalling pathway is an ancient and highly conserved pathway common to all metazoans [1]. It is activated through the binding of glycolipoprotein Wnt ligands to their cognate Frizzled (FZD) receptors and their co-receptors, low-density lipoprotein receptor-related protein 5/6 (LRP5/6) [2]. In the absence of Wnt ligand, cytoplasmic β-catenin, a key transducer of the Wnt/β-catenin signal, is marked for degradation by the destruction complex which is composed of the scaffold protein Axin, adenomatous polyposis coli (APC), casein kinase 1 (CK1) and glycogen synthase kinase 3β (GSK3β) [2,3]. The latter two kinases work together to phosphorylate β-catenin. CK1 primes, while GSK3β phosphorylates serine/threonine residues contained within the N-terminal domain of β-catenin, targeting it for ubiquitination and subsequent proteasomal degradation [2,4]. Activation of the Wnt/β-catenin pathway allows for the recruitment of the scaffold protein Dishevelled (DVL) leading to the disaggregation of the destruction complex and nuclear translocation of cytoplasmic β-catenin and its association with T-cell factor/lymphoid enhancer-binding factor (TCF/LEF) transcription factors, amongst others, to activate the transcription of Wnt/β-catenin target genes [2] (Figure 1). 

Wnt/β-catenin signalling plays a key role in embryogenesis, as evidenced by loss of function gene knockout studies targeting components of the Wnt/β-catenin pathway including LRP5/6 [5], Wnts [6] and FZDs [7] (for review see [8]). The Wnt/β-catenin signalling pathways plays a central role in regulating proliferation and differentiation during development as well as in stem cell renewal [9,10]. These properties make it an attractive prospect for those seeking druggable targets in the treatment of cancer, in which Wnt/β-catenin signalling is often implicated [3]. Manipulation of the pathway is a potentially useful tool for those seeking to generate differentiated cell types in vitro (e.g., hepatocytes) for the regeneration or replacement of diseased organs and in the development of stem cell derived hepatocytes for drug toxicity screening [11]. It is also increasingly well recognised as an important player during liver homeostasis and repair with recent studies supporting its role in proliferation throughout the liver parenchyma [12,13,14,15].

In addition to its signalling function, β-catenin has a structural role in cell adhesion. It was first isolated as a chain with α-catenin and γ-catenin attached to the Ca^2+^-dependent cell adhesion molecule E-cadherin [16,17]. Through its link with E-cadherin, β-catenin forms a physical association between cytoskeleton and transmembrane proteins. This association is crucial during embryogenesis since epithelial cell migration is mediated by the dynamic cadherin-catenin-actin complex [18] (for reviews see [19,20]). 

The adult liver is the largest internal organ and performs numerous functions including energy and nutrient homeostasis, xenobiotic and pathogen elimination, bile acid production and protein secretion [21,22]. The ability of the liver to carry out multiple processes simultaneously is due, in part, to the remarkable phenomenon of hepatocyte heterogeneity. This feature was first noted following observations of variation in the activity of key rate-limiting enzymes and subcellular morphology in hepatocytes located at different coordinates across the lobule, the structural unit of the liver (Figure 2) [23,24,25,26]. The idea was subsequently canonized in the hepatologist’s lexicon through a series of studies on ”metabolic zonation” by Jungermann, Gebhardt, Kietzmann and colleagues [25,27,28,29,30,31,32,33]. Their and others’ work revealed a carefully co-ordinated system which orchestrated labour to maximise functional output to meet both exogenous energy demands and endogenous metabolic demands. The finely tuned integration of multiple interdependent processes serves to minimise redundancy, substrate competition and futile cycling through complementarity and spatial separation. In recent years, the role of the Wnt/β-catenin pathway in the regulation of this arrangement has been scrutinised [34,35,36].

The liver has evolved the capacity to regenerate large quantities of tissue mass following injury. Following a 2/3 partial hepatectomy (PHx), rat livers are able regenerate to full size in just a few days [9,34,35]. The Wnt/β-catenin pathway has been implicated in this process with components of the pathway, including β-catenin and APC, showing a marked upregulation in response to PHx in rats [36]. A comprehensive description of the forces underpinning this ability, as well as those involved in healthy homeostatic renewal, represents an ongoing challenge in liver biology. Numerous theories have been proposed to describe the source of the nascent hepatocytes both in the context of homeostatic renewal and regeneration following injury. The subject of this review will focus on exploring the role of Wnt/β-catenin signalling in orchestrating hepatic development and homeostasis. We contextualise recent experimental findings within the literature which challenge the idea that endothelial derived Wnt/β-catenin signalling is at the heart of homeostatic renewal [37].

### 1.1. Role of Wnt/β-Catenin in Gastrulation

The liver is derived from the multipotent precursor cells of the definitive endoderm (DE). Fate mapping studies in the mouse embryo have indicated the presence of mesendoderm precursors that give rise to the endoderm at approximately embryonic day (E) 7.5. In vertebrates, the Wnt/β-catenin pathway participates by enhancing levels of Nodal growth differentiation factor (Nodal) expression, whose activity is necessary to induce formation of the DE [38]. However, by the end of gastrulation, suppression of Wnt/β-catenin in the anterior endoderm is required for foregut induction and subsequent liver development [39].

The Wnt/β-catenin signalling pathway is involved in the formation of the primitive streak and anterior–posterior axis patterning through regulation of downstream Nodal expression. Nodal ligands belong to the transforming growth factors β (TGFβ) family of secreted growth factors and signal via the ALK/SMAD cascade. During gastrulation, Nodal signalling plays a key role in the induction of mesendoderm specification and enhances the expression of GATA-binding factor 1 (GATA1), Mix-like, Sox and transcription factor forkhead box (FOX) transcription factors [39,40]. Depending on the level of Nodal signalling, either endoderm or mesoderm is specified. Over time, high levels of Nodal signalling result in endoderm specification, whereas low doses result in mesoderm specification [39]. Through this mechanism, Wnt/β-catenin signalling via β-catenin and TCF/LEF transcription factors can act to induce high levels of NODAL gene transcription, thus promoting endoderm specification. Applied in an in vitro setting, it has been shown that pre-exposure of human embryonic stem (hES) cells to Wnt/β-catenin signals is required in order for these cells to competently respond to Activin/Nodal signalling. This is a pre-requisite for the homogenous formation of DE expressing the pan-endoderm markers C-X-C motif chemokine receptor 4 (CXCR4) and FOXA2 [41]. 

In mice, Nodal protein expression in the primitive streak is controlled by Wnt3 signalling through TCF/LEF DNA-binding sites in the NODAL gene promoter [42,43]. Likewise, in zebrafish and *Xenopus*, a maternal Wnt/β-catenin signal appears to induce high Nodal expression levels in the dorsal-anterior mesendoderm via interactions with VegT and signals from the maternal yolk syncytial layer (YSL) [42,43]. The YSL is an extra-embryonic syncytial tissue, whose function is critical for endoderm formation [44]. During gastrulation, the Wnt/β-catenin pathway generates a gradient of Nodal signalling in the primitive node and temporally regulated expression profile within the streak. These actions result in a higher concentration of Nodal in the anterior region during early streak formation, which helps to induce the DE and maintain sustained Nodal activity in cells that migrate from this region [45]. Moreover, a network of transcription factors required for DE development is established by Nodal signalling. This includes activation of transcription factors, including FOXA2 and SRY-related HMG-box (SOX) 17, which are required for specification, commitment and future patterning of the DE [46,47].

### 1.2. Gut Tube Patterning

Following gastrulation, the DE forms into the primitive gut tube along the anterior–posterior axis from which the foregut, midgut and hindgut domains emerge (Figure 3). Regulation of Nodal by Wnt/β-catenin signalling is important in initial endoderm patterning. In *Xenopus* and zebrafish, activation of Wnt/β-catenin signalling on the future dorsal side of the blastula regulates Nodal signalling levels to induce expression of the haematopoietically expressed homeobox (HHEX) gene which is required for foregut development in the anterior endoderm [48,49]. Expression of downstream transcription factors including HHEX and FOXA2 is induced, and they are predominantly expressed in the anterior aspect of the embryo. These transcription factors promote foregut endoderm fate and future liver development [38,46]. Signals from adjacent mesoderm tissue, including Wnts, fibroblast growth factor (FGF) 4 and bone morphogenic protein (BMP), repress foregut development by blocking the expression of HHEX and FOXA2 [16,50]. Therefore, foregut formation only occurs when FGF4 and Wnt/β-catenin signalling is supressed in the anterior endoderm [16,50]. Here, secreted FZD-related protein 5 (SFRP5), a secreted Wnt inhibitor, is necessary to maintain the foregut progenitors, and subsequent liver development [51]. Thus, SFRP5 may be involved in determining fate specification of cells destined to form organs of the gastro-intestinal (GI) tract by modulating Wnt/β-catenin signals [52]. Consistent with this concept, McLin et al. found that the foregut did not develop in the anterior endoderm when Wnt/β-catenin was activated [50]. However, inhibition of Wnt/β-catenin signalling, through the overexpression of the destruction complex member GSK3β, was sufficient to induce foregut fate, highlighting the importance of this pathway in endoderm patterning [50]. 

In contrast to the requirement of Wnt/β-catenin signalling suppression for foregut development, activation of the pathway is required later in development for hepatic induction and liver bud growth [53]. Graded signals derived from the cardiac mesoderm and the lateral plate mesoderm, such as FGF and BMP, are required for hepatic specification [54,55]. In zebrafish and *Xenopus*, Wnt ligands secreted from mesoderm cells act by promoting the expression of HHEX and prospero homeobox 1 (PROX1), which constitute two of the earliest liver-specific genes [56,57]. Notably, a study in zebrafish embryos demonstrated a role for Wnt2bb, a homolog of Wnt2b, during hepatic specification. Defects in liver formation and expression of early liver markers (such as HHEX and PROX1) were observed when Wnt2bb was absent in these embryos [58], indicating a positive role for this pathway in liver specification. In mice, FGF and BMP are required for hepatic induction; however, Wnt/β-catenin signalling is not involved in this process [50,57]. 

Following hepatic specification, liver bud development begins accompanied by the proliferation and expansion of the hepatic progenitor population. Wnts and other signalling molecules, including FGF and hepatocyte growth factor (HGF), are predominantly responsible for this process [59]. Tan et al. demonstrated the importance of Wnt/β-catenin signalling in regulating hepatoblast proliferation and increasing transcription of CyclinD1 in transgenic mice [60]. Moreover, FGF has been reported to increase hepatoblast proliferation by activating the β-catenin cascade via FGFR2b [61]. Lastly, HGF activity depends on its cognate receptor Met, which also interacts with and activates the β-catenin cascade to induce hepatoblast proliferation [62]. These findings highlight the importance of studying the implications of high HGF levels and t association with hepatocellular cancer development.

### 1.3. Differentiation of Hepatoblasts into Hepatocytes and Cholangiocytes

Induced by signals derived from the mesoderm, bipotent hepatoblasts in the liver bud can differentiate into either hepatocytes or cholangiocytes [59]. Wnt/β-catenin activity promotes differentiation of hepatoblasts to a cholangiocyte rather than hepatocyte lineage. Wnt3a treatment in mouse liver explants enhances cholangiocyte formation when compared to control cultures [63]. Furthermore, Cordi et al. demonstrated the importance of the Wnt/β-catenin in cholangiocyte differentiation by supressing β-catenin in transgenic mice, via a loss-of-function approach [64]. Results indicated that β-catenin activity was required for differentiation of hepatoblasts to cholangiocyte precursors, as the number of hepatocyte nuclear factor 4 alpha (HNF4α)-positive cells was reduced when β-catenin was inhibited. However, overexpression β-catenin perturbed cholangiocyte differentiation, suggesting that β-catenin activity must normally be finely controlled [64]. 

### 1.4. Role of Wnt in Directed Differentiation of Pluripotent Stem Cells to Hepatocyte-Like Cells

Due to the dual properties of being able to divide without limit and ultimately differentiate into any cell type of the body, pluripotent stem cells (PSCs) represent a powerful tool in the generation of differentiated cell types [65]. Generating physiologically relevant cells that reproduce the characteristics of adult primary hepatocytes could be of enormous value as a source of cells for transplantation into patients with liver failure as well as in vitro for evaluating drug metabolism and toxicity and modelling of liver disease [66].

The Wnt/β-catenin signalling pathway is an active player during embryonic liver development and is characterized by a strictly temporally regulated profile [67]. We can apply this knowledge to the development and refining of protocols for the directed differentiation of PSCs to hepatocyte-like cells (HLCs). Though there is some variation between individual protocols, most protocols for the directed differentiation of PSCs to HLCs involve three steps: induction of DE, hepatoblast differentiation and HLC maturation. Each stage involves treatment with exogenous growth factors and cytokines [68]. Wnt/β-catenin signals are vital to the generation of DE and hepatoblast differentiation stages. 

In 2008, Hay et al. reported that the combination of Activin A and Wnt3a efficiently induced differentiation towards a hepatic endoderm phenotype and generated viable HLCs [69]. As an alternative to Wnt3a, the use of small molecule GSK3β inhibitors, such as CHIR9021, to activate the Wnt/β-catenin signalling pathway during endoderm specification has proved to be effective when administered transiently on the first day of endoderm generation [70]. Furthermore, Bone et al. reported a highly potent and selective GSK3β inhibitor (termed 1 m) with the capacity to generate DE cells from human embryonic stem cell (hESC) colonies [71]. Later, Chen et al. refined the protocol to include HGF. Using HGF in combination with Activin A and Wnt3a enhanced the expression of DE markers FOXA2 and SOX17, but also reduced the time it took to differentiate into HLCs [72]. Interestingly, excessive Wnt3a treatment alone does not efficiently induce HLCs [70]. Touboul et al. reported that inhibition of the Wnt/β-catenin pathway is required to promote hepatoblast differentiation to hepatocytes. Their findings suggest that Wnt3a treatment should be strictly temporally regulated in order to generate endodermal cells with high hepatocyte differentiation capacity. With the idea of mimicking the role of Wnt/β-catenin signalling during normal liver development, temporal activation of Wnt/β-catenin signalling was required for induction of the DE, whereas its inhibition allowed specification of the posterior foregut and then HLCs. However, Touboul et al. reported that consecutive activation of the Wnt/β-catenin pathway promoted the production of proliferative bipotent hepatoblasts, which then improved maturation of the cells with increased expression of adult hepatic genes, such as Albumin and cytokeratin 18 (CK18) [25,73].

### 1.5. The Role of Wnt/β-Catenin Signalling in Metabolic Zonation of Adult Liver

Metabolic zonation is typically associated with functions served by parenchymal cells, which in the case of the liver consist of hepatocytes and comprise 80% of the liver mass and 60% of the liver’s cells [72]. At ~20 hepatocytes long in humans, the acinus is the smallest functional unit of the liver. Several acini connect to form the smallest structural unit of the liver, the lobule. The lobules form a repeating hexagonal pattern with central veins at each centre and bile ducts, portal veins and hepatic arteries clustered at each vertex (collectively termed portal triads) (Figure 2) [74]. The liver is highly vascularised receiving O_2_ rich blood from the hepatic artery and nutrient rich blood from the portal vein. This dual blood supply speaks to the organ’s function as a general-purpose filter and nutrient sensor. Blood entering the liver flows through sinusoidal capillaries carrying nutrients and toxins from the GI tract. Within the sinusoids, blood is processed hierarchically by cords of polarised hepatocytes which line the space of Disse, a gap which separates the hepatocytes from the sinusoidal epithelium. Microvilli projecting from hepatocyte basal membranes occupy this space facilitating exchange of nutrients, metabolites and O_2_ with the blood through fenestrae in the sinusoidal endothelium. This lobular structure produces concentration gradients of metabolites, hormones, proteins, O_2_ and by-products of metabolism (e.g., carbon dioxide (CO_2_)) along the porto-venous axis. Hepatocytes towards the afferent (periportal) end are, therefore, exposed to a microenvironment distinct from that at the efferent (perivenous) end. This translates to gradients of function across the porto-venous axis. For example, hepatocytes in the periportal zone exhibit activity associated with higher oxidative requirements including glycogen synthesis and storage, amino acid utilization, ammonia detoxification and beta-oxidation, while hepatocytes in the perivenous zone perform non-oxidative functions including biotransformation reactions, glutamine synthesis and glycolysis [25,73] (Figure 2). Analysis using a database of known molecular pathways and associated functions (KEGG) confirmed that components of the oxidative phosphorylation pathway are more highly expressed in periportal hepatocytes where pO_2_ is higher (65 mmHg) than in perivenous hepatocytes (30 mmHg) [73].

### 1.6. The Role of the Wnt/β-Catenin Pathway in Metabolic Zonation of Liver 

Several mechanisms have been proposed to account for the organisation of hepatocytes into metabolically distinct zones including O_2_ or nutrient gradients [28,30,75,76], differences in composition of the extra-cellular matrix (ECM) along the porto-venous axis [32] and secreted endothelial Wnt ligands [37,77,78,79,80,81]. That each of these contributes to overall patterning of liver zonation is conceived under the umbrella theory of postdifferentiation patterning. This broadly states that the microenvironment created by liver architecture, gradients of blood constituents and milieu of morphogens conspire to produce metabolic zonation [82]. 

In recent years, the Wnt/β-catenin signalling pathway has been shown to be the key mediator of liver zonation within the context of ammonia detoxification [83,84,85]. Briefly, ammonia is generated predominantly from two sources: as a by-product of amino acid metabolism and as a waste product from gut bacteria. The liver contains two systems for the removal of ammonia: the urea cycle and the enzyme GS. Interestingly, these systems are expressed in a mutually exclusive pattern across the liver acinus [86,87]. A sharp demarcation is observed with GS expression being limited to the last 1–2 layers of perivenous hepatocytes, whereas the urea cycle enzyme CPS1 is expressed in all hepatocytes of the periportal, intermediate and first few layers of the perivenous zone (Figure 2). The stable expression of these two systems led to it becoming a model of zonation in liver biology. However, the mechanism responsible for the distribution of these two enzymatic systems remained unknown until relatively recently. 

In 2002, three independent groups noted that active nuclear β-catenin signalling correlated with a significant increase in GS expression [88,89,90]. Further evidence came from CATNB-mutated hepatomas in which unphosphorylated β-catenin is constitutively activated. These studies revealed elevated transcription levels of genes associated with the perivenous zone under β-catenin control including GS, Cytochromes P450 (CYP) 2E1, CYP2C, coxsackievirus and adenovirus receptor (CAR) and glutathione S-transferase (GST) [81]. In a seminal study in 2006, Benhamouche et al. demonstrated that the Wnt/β-catenin pathway underpins zonation of ammonia detoxification in the liver, providing spatial separation of GS and urea cycle enzymes. Based on data showing the nuclear β-catenin dependent upregulation of glutamine synthesis pathway enzymes (GS, ornithine aminotransferase (OAT) and glutamate transporter 1 (GLT1)) and increased β-catenin activity in APC^−/−^ mice, the group hypothesised that APC suppressed β-catenin activity. They suggested that this interaction could give rise to the mutually exclusive expression of ammonia detoxification systems in the liver [84,89,91]. Conditional loss of APC-induced expression of 32 genes including four known perivenous zone genes: GS, GLT1, OAT and arginine vasopressin receptor 1A (AVPR1A). Of 31 suppressed genes, two were identified as periportal zone genes: Glutaminase 2 (GLS2) and Arginase 1 (ARG1), as determined by microarray and quantitative real time-PCR (q-PCR). Blockade of Wnt/β-catenin signalling with the Wnt antagonist DKK1, a negative regulator of β-catenin signalling, resulted in the reduced expression of perivenous genes including Rh family B glycoprotein (RHBG), leukocyte cell-derived chemotaxin 2 (LECT2), Axin2, OAT, GLT1 and AVPR1A. In contrast, some periportal genes were upregulated across the lobule with the exception of the most distal perivenous layer, including ARG1, GLS2 and phosphoenolpyruvate carboxykinase (PEPCK) [84].

Studies in intestine previously demonstrated that the oncogene c-Myc was a downstream effector of Wnt/β-catenin signalling [92]. The question arose of whether c-Myc plays a similar role in liver zonation and whether conditional deletion of APC and c-Myc would replicate Benhamouche’s findings [85]. Deletion of APC led to rapid deterioration in the health of the mice due to acute ammonia toxicity and, in accordance with Benhamouche, upregulation of perivenous genes across the lobule including GS, GLT1 and RNase4. Periportal genes including CPS1, GLS and PEPCK were concomitantly downregulated at both the protein and RNA level. β-catenin RNA itself, however, was not affected. Deletion of c-Myc on a background of APC deletion did not rescue the phenotype indicating that the Wnt/β-catenin pathway regulates zonation via a c-Myc-independent mechanism. Conditional knockout of β-catenin resulted in a loss of perivenous GS expression, confirming the role of Wnt/β-catenin signalling in ammonia detoxification zonation [79]. Many studies have since confirmed and expanded these findings [77,78,83].

In 2017, Halpern and colleagues developed a map of gene expression in the liver using single cell RNA sequencing (scRNASeq) and single molecule fluorescence in situ hybridisation. The authors confirmed previous reports that hepatocytes were highly heterogenous and exhibited clear gradients of function. Additionally, they found that a remarkable 50% of liver-associated genes display significant hepatic zonation (3496 of a total 7277 genes). Of these, some genes were expressed most highly in the intermediate zone such as hepcidin antimicrobial peptide (HAMP), HAMP2, IGF binding protein 2 (IGFBP2), major urinary protein 3 (MUP3) and CYP8b1. They also found that many perivenous genes were upregulated in APC^−/−^ mice (810 of 3496 zonated genes), in which Wnt/β-catenin signalling is constitutively activated. In contrast, periportally-associated genes were downregulated (196 of 3496 zonated genes). Under hypoxic conditions, 95 genes were upregulated in the perivenous area, while 45 were downregulated suggesting a role for pO_2_ in zonation [73].

A second study by Halpern and colleagues in 2018 focused on the possibility of zonation of non-parenchymal cells (NPCs) in the liver. They identified several distinct cell types based on enriched markers for endothelial cells, T-cells, dendritic cells, Kupffer cells, liver capsule macrophages and neutrophils. By using a paired sequencing approach based on the previously established hepatocyte gene expression map, they were able to distinguish distinct populations of NPCs at different locations within the liver lobule [73,93]. In total, 40% of cells that make up the liver are NPCs, and of these, as many as 50% are liver endothelial cells (LECs) [72,94]. Though previous studies had found heterogeneity in LECs in terms of numbers and morphology [29], Halpern was able to confirm, using sensitive high throughput techniques, that ~35% of LEC-associated genes were zonated [93]. Interestingly, they found that a distinct subset of perivenous LECs, those which comprise the central vein, expressed and secreted the Wnt antagonist DKK3. This was in contrast to adjacent central vein LECs which expressed Wnt/β-catenin agonists Wnt2, Wnt9b and R-spondin (RSPO) 3 in accordance with a model of zonation governed by centrilobular Wnt/β-catenin signalling. The question arises: why do some LECs in the central vein express the antagonist and others express the agonists? The authors postulated that this perhaps constitutes part of a negative regulatory system which inhibits the oncogenic activity of the Wnt/β-catenin pathway. Furthermore, they found differences in LECs of the central vein and sinusoidal LECs of the perivenous zone. Central vein LECs expressed more RSPO3 and the cell adhesion protein Cadherin (CDH) 13, whereas sinusoidal LECs expressed Wnt2 and the proto-oncogene and cytokine receptor KIT [93]. Examining the relative contribution of central vein versus sinusoidal LECs to zonation may yield new insights into the fine tuning of zonation. 

LECs comprise the central and portal veins and sinusoids and are known to regulate immune responses and secrete morphogens [93]. Endothelial cells of the portal vein express and secrete DKK2, a negative regulator of Wnt/β-catenin signalling, whereas LECs of the central vein and perivenous sinusoid secrete Wnt2, Wnt9b and RSPO 1, 2 and 3 [12,37,78,83]. RSPO signals act as a rheostat system by signalling through leucine rich repeat containing G-protein coupled receptors (LGR) 4-6 and potentiating endothelial derived Wnt/β-catenin signalling. This is achieved by promoting the turnover of the ring finger protein (RNF)43/zinc and ring finger (ZNRF) 3 receptor through RSPO binding to and activation of the LGR4 receptor which triggers endocytosis of the RNF43/ZNF3 receptor, thus preventing RNF43/ZNF3-mediated turnover of FZD receptors [78,95]. The accumulated evidence of two decades of research has built a convincing picture of the morphogens which govern the zonation of ammonia detoxification systems, namely Wnt ligands, secreted DKK proteins and RSPOs.

While much is now known of the fundamental role of Wnt/β-catenin signalling in liver zonation, questions remain around how zonation arises in the developing liver, how it is re-established following injury and to what extent Wnt/β-catenin signalling maintains zonation of other metabolic functions (e.g., ketogenesis, lipogenesis and gluconeogenesis) and non-metabolic proteins such as those involved cell-cell adherence. Ma et al. recently addressed some of these questions and extended the influence of Wnt/β-catenin signalling to include zonation of the adhesion protein E-cadherin and tight junction component Claudin2. GS and CPS1 are first expressed in the developing embryo at E15.5 and E13.5, respectively, with GS expression coinciding with the upregulation of nuclear β-catenin [77,96]. Claudin2 was not found to be expressed in hepatocytes at E18.5, only becoming broadly expressed across the lobule at two days postpartum before showing a more restricted perivenous expression pattern at postpartum day 15. E-cadherin, a periportal marker, was found to be broadly expressed at E18.5 and postpartum day two, though did not adopt its predominantly periportal pattern until postpartum day 10 [83,95]. 

Using lineage tracing of sinusoidal LECs (as determined by expression of the sinusoidal LEC marker lymphatic vessel endothelial hyaluronan receptor 1 (LYVE-1)) the authors were able to determine that sinusoidal LECs adjacent to the central vein expressed Wnt2, Wnt9b and RSPO3 becoming limited to expression of Wnt2 alone more distally from the central vein. Furthermore, they demonstrated that ablation of sinusoidal LECs had little impact on initiation of Wnt/β-catenin dependent zonation but was involved in the maintenance of zonation in adulthood. The authors examined Claudin2 expression following CCL_4_-mediated ablation of perivenous hepatocytes. In accordance with a recent study which found evidence for the role of Wnt/β-catenin in regeneration following CCL_4_-mediated injury [97], they demonstrated that endothelial Wnts are required for reinstating Claudin2 zonation. Interestingly, they found that E-cadherin expression was transiently expanded to the perivenous region following CCL_4_ treatment and present in the intermediate zone at 2–5 days post-treatment, before resuming its periportal expression during the repair phase [83]. Conditional deletion of endothelial Wnt ligand secretion protein (WLS), a protein which facilitates the exocytosis of Wnt ligands, inhibits expression of GS, CYP2E1 and Claudin2 in perivenous hepatocytes in adult mice [83]. Together these findings suggest that during development, Claudin2 does not share the tightly Wnt/β-catenin regulated expression of GS, though it does rely on Wnt/β-catenin signalling for maintenance in adulthood. Further studies are needed to build a clearer picture of how zonated genes are regulated during development.

### 1.7. Homeostatic Renewal and Regeneration of the Liver

Recent studies have agreed on the conclusion that it is resident hepatocytes themselves which are the major locus of hepatocyte neogenesis and reservoir of potential plasticity in homeostatic renewal and in response to injury [12,13,14,15,98]. Early research had suggested that the liver parenchyma was repopulated, both in human and murine models, by bi-potent progenitor cells termed “oval cells” resident in the terminal bile ducts [99,100,101]. Like hepatoblasts, these bipotent cells were thought to regenerate both the parenchyma and biliary tissue in response to toxin-induced injury [100,102]. Subsequent research demonstrated that hepatocytes could repopulate biliary cell mass in response to bile duct ligation injury models through transdifferentiation [103] and following PHx through hepatocyte proliferation and hypertrophy [104]. Plasticity and the notion of transdifferentiation are increasingly accepted features of liver biology. SOX9, a stem cell marker, is constitutively expressed at low levels by periportal hepatocytes and bile duct cholangiocytes. Using lineage tracing, this population was shown to upregulate proliferation to regenerate lost liver mass in response to acute but not chronic CCL_4_-induced injury [105]. Hepatocytes have also been induced to generate de novo biliary tissue through TGFβ-mediated transdifferentiation [106]. Recently Lin et al. demonstrated that distributed hepatocytes expressing the stem cell marker telomerase reverse transcriptase (telomerase) were disproportionately responsible for hepatocyte proliferation [15]. While there is growing agreement that distributed hepatocytes are responsible for proliferation, the relative contribution of telomerase expressing hepatocytes specifically is yet to be confirmed.

Wnt/β-catenin signalling confers a proliferative capability in the liver throughout development and adulthood by promoting transcription of genes associated with re-entry into the cell cycle including CyclinD1 and c-Myc [67,107,108]. Resident stem cells from many tissues, for example intestine, express components of the Wnt/β-catenin signalling machinery LGR5 and Axin2, and Wnt/β-catenin signalling has been shown to maintain self-renewal in stem cells [10,109,110]. Evidence from tumorigenesis studies support the idea that active β-catenin signalling promotes hepatocyte proliferation [91,111,112]. In 2015, Nusse and colleagues hypothesised that locally secreted endothelial Wnts signal to adjacent hepatocytes to support a proliferative capability in this Axin2 expressing subpopulation (Figure 1). They sought to decipher the role of Wnt/β-catenin signalling in a landmark study in mice. Using lineage tracing of tamoxifen inducible Axin2-GFP as a surrogate reporter of Wnt/β-catenin activity, the authors traced the lineage of perivenous hepatocytes over 365 days. The Axin2 gene contains WREs in the promoter region making it a direct indicator of Wnt/β-catenin activity. Through pulse labelling of Axin2 positive cells with permanent GFP expression over a one-year time course, they concluded that Wnt^+^/Axin2^+^ hepatocytes adjacent to the central vein were the primary source of hepatocyte neogenesis during homeostatic renewal. New cells appeared to emanate from the perivenous region eventually populating the entire lobule [37]. This hypothesis implied a model of hepatocyte differentiation that would place the relatively plastic and Wnt/β-catenin active perivenous hepatocytes higher in Waddington’s landscape than periportal hepatocytes [113], implying that the life of a hepatocyte involves multiple identities as it migrates against the flow of blood to the portal vein. In effect, they suggested a model akin to the streaming liver theory in reverse [114].

A string of recent studies proffers a robust rebuttal to Nusse’s conclusion that homeostatic renewal occurs predominantly in the perivenous region with hepatocytes migrating along the acinus over time, eventually repopulating the entire lobule. Nusse’s study utilised a CreERT2 cassette to disrupt one Axin2 allele. Axin2 is both a downstream target of Wnt/β-catenin signalling as well as negative regulator, thus producing a negative feedback loop [115]. It has been suggested that possible Axin2 haploinsufficiency resulting from the heterozygous deletion of an Axin2 allele may have granted a proliferative advantage to labelled cells through dysregulation of the negative feedback loop, therefore inflating the number of cells derived from this population [12,13]. In a comparable study, Lin et al. traced the lineage of telomerase expressing hepatocytes by crossing Tert^CreERT2/+^ mice with a Rosa26^LSL-Tomato/+^ reporter strain to permanently label hepatocytes expressing telomerase and their progeny. In this study, they found that hepatocyte proliferation was predominantly restricted to a sparse yet uniformly distributed telomerase expressing a hepatocyte subpopulation and that this represented a stem cell-like proliferative niche. Hepatocytes expressing telomerase were observed to disproportionately repopulate the liver producing clonal clusters comprising an average 6–10 cells at 1 year [15]. Chen et al., however, have since reported a reduced clonal cluster size of just 1.1 on average after 13.6–13.7 months in a separate lineage tracing study [13]. In light of the fact that enhanced telomerase expression promotes proliferation, the possibility that Tert^CreERT2/+^ Rosa26^LSL-Tomato/+^ transgenic mice may bias telomerase expressing cells towards proliferation must be ruled out [13,116]. Where Wang et al. and Lin et al. found evidence for discrete subpopulations with enhanced proliferative potential, more recent experiments have found proliferation to be more distributed and not significantly limited to a particular subpopulation or location [12,13,14]. In direct contrast to the Wang et al. finding, an inducible knockout of perivenous hepatocytes using diphtheria toxin A (DTA) in combination with lineage tracing of Axin2 without the accompanying heterozygous deletion of an Axin2 allele, demonstrated upregulation of Axin2 across the lobule. These cells were found to be capable of restoring the perivenous niche [12]. Furthermore, Chen et al. found through random labelling of hepatocytes with q Rosa26-Rainbow Cre reporter that clonal expansion of hepatocytes did not occur predominantly in the perivenous region, rather it was distributed across the lobule with a slight increase in proliferation in the midlobular zone in accordance with Lin et al. [13,15]. There is also consensus around proliferation in response to injury. Adjacent hepatocytes respond with proliferation in response to local acute injury, whereas in the case of chronic injury, a wider upregulation of proliferation is seen [12,13,14]. Interestingly, Sun et al. found that Axin2 was upregulated in response to injury across the lobule, suggesting that Wnt/β-catenin signalling may be facultatively employed to regenerate liver parenchyma.

### 1.8. Wnt/β-Catenin Crosstalk with Hh and HIF Pathways 

All intracellular pathways are inextricably linked with the activity of other pathways. This interdependence allows for the formation of the dynamic physical infrastructure which enables cells to communicate with each other and with their environment. Within the context of Wnt/β-catenin signalling in the liver, two pathways are of particular interest: (i) the Hh pathway and (ii) the HIF pathway [117]. In this section, we briefly highlight insights into how interactions with these pathways help regulate liver homeostasis [118,119]. 

Like the Wnt/β-catenin pathway, the Hh pathway is important both in embryological development controlling growth, tissue patterning, mitogenesis as well as adult homeostasis and tissue repair in vertebrates [112,114,115]. In mice, conditional homozygous deletion of SMO, an Hh signalling rate limiting protein, in the liver negatively impacts development, producing smaller offspring with perturbations to insulin and glucose homeostasis due to disrupted IGF1 and IGFBP1 signalling [116]. Hh ligands Sonic hedgehog (Shh) and Indian Hedgehog (Ihh) are locally secreted in the liver from endothelial cells, cholangiocytes and stellate cells in which Hh signalling is most active [112,114]. Hh signals form local concentration gradients differentially impacting gene transcription cascades as a function of distance from the Hh signal [115]. Crosstalk between Hh and Wnt/β-catenin signalling has long been known to occur in the liver and has been explored within the context of hepatic cancers [117,118]. In light of the finding that, in vitro, two negative regulators of Wnt/β-catenin signalling, Wnt5a and DKK1, were upregulated in response to Hh signalling [118], it was proposed that Wnt/β-catenin and Hh pathways may interact to govern hepatocyte zonation [83]. An OMICS approach was taken to determine the spatial distribution and points of interactions between the two pathways. Wnt/β-catenin and Hh signalling machinery were found to be heterogeneously and inversely distributed within the liver. The RSPO-LGR4/5-ZNRF3/RNF43 module (LGR5, RSPO1-3, RNF4) and Wnt/β-catenin (Wnt5a, Wnt9b) signalling component expression were elevated in the perivenous region and lower in the periportal region, while the Hh signalling proteins Shh and SUFU (a negative regulator of Wnt/β-catenin signalling [119]) were found to be preferentially expressed in the periportal zone. Ihh ligands were one exception to this rule, observed to colocalise with GS and Axin2 in perivenous hepatocytes. Activation of Wnt/β-catenin signalling results in concomitant Hh signalling, and ablation of the Hh signal using an inducible SMO knockout model results in upregulation of the RSPO-LGR4/5-ZNRF3/RNF43 module. These findings imply that Hh signalling in both hepatocytes and NPCs, which is normally supressed in healthy adult tissues and hepatocytes, not only plays an important role in liver zonation and displays a zonated pattern but is complimentary to Wnt/β-catenin signalling. 

HIF signalling has long been hypothesised to play an instructive role in liver zonation [25]. Evidence for crosstalk between Wnt/β-catenin and HIF signalling has consistently drawn a link between hypoxia and an increase in Wnt/β-catenin activity. The concentration of O_2_ in the perivenous region is approximately half that of the periportal zone dropping from 65 to 30mmHg due to the oxidative intensive functions of periportal hepatocytes [76]. All three HIFs (HIF1, HIF2, HIF3) show a predominantly perivenous zonation pattern and are activated through phosphorylation by VHL protein and one of three similarly zonated HIF prolyl 4-hydroxylases [27,120]. Hypoxic conditions (10% O_2_) have been shown to upregulate genes under β-catenin control including CyclinD1 and c-Myc [121], while knocking out ARNT and HIF proteins produces a reduction in β-catenin transcriptional co-factor TFC/LEF in stem cells [122]. In 2007, Kaidi et al. demonstrated how β-catenin and HIF interact using DNA-protein and protein–protein interaction techniques. HIF1α competes with TCF/LEF for direct binding to β-catenin, and the HIF1α/β-catenin complex interacts with WREs in the promoter regions of HIF-1 target genes including CA-IX, Glut1, VEGF and COX-2. Furthermore, they demonstrated that β-catenin can potentiate HIF-1-mediated transcription, enhancing the response to hypoxia [122,123]. One way in which Wnt/β-catenin and HIF pathways may interact is through HIF1 binding to HREs in the promoter region of the β-catenin transcriptional coactivator BCL9 (Figure 1). This interaction synergistically promoted Wnt/β-catenin signalling under hypoxic conditions [119]. Collectively, these studies provide convincing evidence that variations in pO_2_ in the liver acinus participates in the regulation of the Wnt/β-catenin/HIF axis. 

## 2. Conclusions

In this review, we have highlighted important developments in the study of Wnt/β-catenin signalling in liver development and homeostasis. The Wnt/β-catenin pathway has a conserved role in liver development where its activity is tightly regulated both spatially and temporally. During gastrulation, high levels of Nodal signalling induced by Wnt/β-catenin signals promote endoderm specification. By the end of gastrulation, suppression of Wnt/β-catenin signalling in the anterior endoderm permits induction of the foregut endoderm. Protocols facilitating the directed differentiation of iPSCs to HLCs reflect this fact with the administration of Wnt/β-catenin activators Wnt3a and Activin A or GSK3β inhibitors CHIR9021 and 1m being restricted to the first three days of culture [70,71]. 

The role of Wnt/β-catenin signalling in the maintenance of the zonation of hepatic parenchyma is now well established. Recent research has also indicated a role in the zonation of NPCs as well as an expansion of the Wnt/β-catenin pathway to incorporate the RSPO-LGR4/5-ZNRF3/RNF43 module [73,78,93,95]. Halpern (2018) found a distinct subpopulation of LECs in the perivenous region. Some expressed Wnt/β-catenin morphogens, whilst others expressed the Wnt/β-catenin negative regulator DKK3. This suggests that the mechanisms governing Wnt/β-catenin signalling, even within the perivenous LEC compartment, may be more complicated than first anticipated. While a large amount is known about the role of Wnt/β-catenin signalling in the developing and adult liver, questions remain around the initiation of zonation where it has been shown that Wnt/β-catenin signalling is not required [37,83].

With respect to Wnt/β-catenin signalling in the context of homeostatic renewal in the liver, the controversy around exactly which cells are responsible for proliferation may be waning with consensus building around a model of proliferation which sees hepatocytes as the source of neogenesis and facultative contributor to regeneration following injury. Wang et al. suggested that perivenous hepatocytes were at least disproportionately responsible for homeostatic renewal of liver parenchyma. However, new research now makes a pan-zonal model of hepatocyte-driven homeostatic renewal seem more likely [12,13,14,15]. Whether this responsibility is shared by all hepatocytes or a select few and to what extent this differs in response to injury remain to be confirmed with some claiming a role for telomerase positive hepatocytes [13]. Together, these findings lend support to the idea that potency in the liver is distributed rather than limited to a compartment of specialised cells in either the perivenous or periportal region. A model of the liver which sees each hepatocyte as a potentially facultative contributor to regeneration governed by the finely tuned activity and crosstalk of multiple pathways (Wnt/β-catenin, Hh, HIF and other signalling pathways (notably Hippo and FGF)) may be considered. 

By continuing to improve our understanding of this ancient and fundamental signalling pathway, we draw closer to a more complete understanding of one of the mechanisms that enabled life to transition from single to multicellular organisms. With the advent of high throughput technologies, advances in data analysis tools and ever greater resolution of gene and protein expression profiles, the integration of all pathways into a unified model must remain a goal for future research.

## Figures and Tables

**Figure 1 genes-11-01163-f001:**
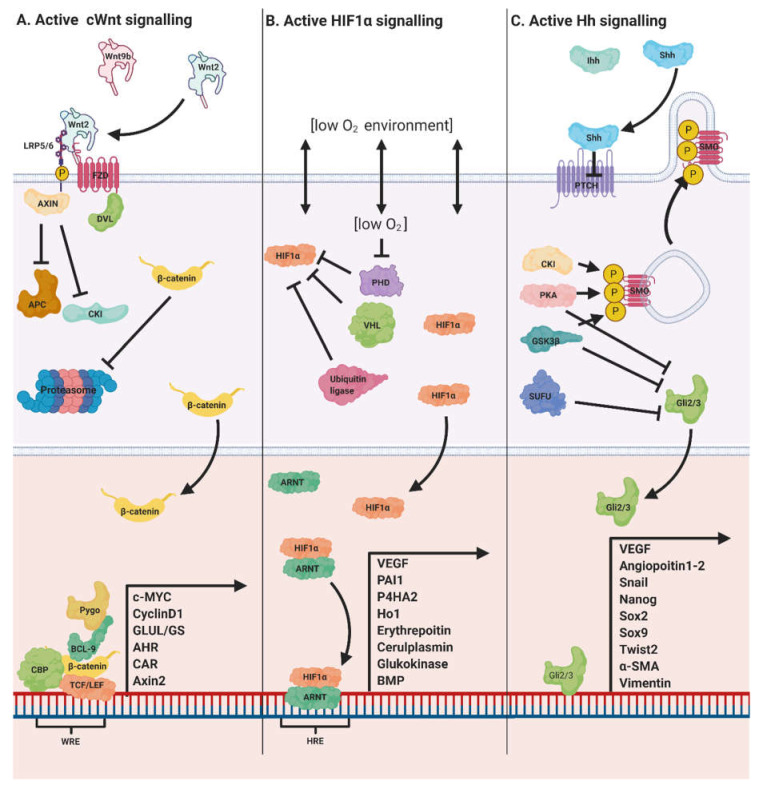
Summary of Wnt/β-catenin, Hedgehog and hypoxia inducible factor (HIF) pathways and their downstream mediators in the liver. (**A**) Active Wnt/β-catenin signalling. Wnt2 and Wnt9b secreted from central vein endothelial cell complex with LRP5/6 and FZD receptors. The β-catenin destruction complex is prevented from assembling allowing cytoplasmic accumulation of β-catenin which in turn permits β-catenin’s translocation to the nucleus. β-catenin then complexes with co-factors including CREB binding protein (CBP), TCF/LEF, B-cell lymphoma 9 (BCL-9) and pygopus protein (Pygo) to promote transcription of genes under β-catenin transcriptional control including c-Myc, CyclinD1, GS, aryl hydrocarbon receptor (AHR) and constitutive androstane receptor (CAR). (**B**) Active HIF1 signalling. Constitutively expressed HIF1α subunits avoid ubiquitin ligase, HIF prolyl hydroxylase (PHD) and VHL (von Hippel Lindau) facilitated proteasomal degradation when PHD is inactivated by low pO_2_. HIF1α subunits are then free to translocate to the nucleus where they dimerise with aryl hydrocarbon receptor nuclear translocator (ARNT) beta subunits to form HIF1 which is then able to bind to HIFresponse elements (HREs) and affect transcription of genes including vascular endothelial growth factor (VEGF), plasminogen activator inhibitor 1 (PAI1), P4HA2, HO1, erythropoietin, ceruloplasmin glucokinase and bone morphogenic protein (BMP). (**C**) Active Hh signalling. Hh ligands bind and inhibit their cognate receptor patched (PTCH). PTCH is no longer able to inhibit the G-protein coupled receptor smoothened (SMO), and SMO is phosphorylated by CKI and GPRK2 causing translocation from intracellular endosomes to the plasma membrane of primary cilia. Gli proteins (Gli1, Gli2, Gli3) can then escape association with suppressor of fused homolog (SUFU) and phosphorylation by GSK3β, protein kinase A (PKA), CKI and resulting in proteasomal degradation. It is then free to translocate to the nucleus to effect transcription of Hh target genes including VEGF, angiopoietin 1 and 2; Snail, Nanog, SOX2 and 9; Twist2, α-SMA and vimentin.

**Figure 2 genes-11-01163-f002:**
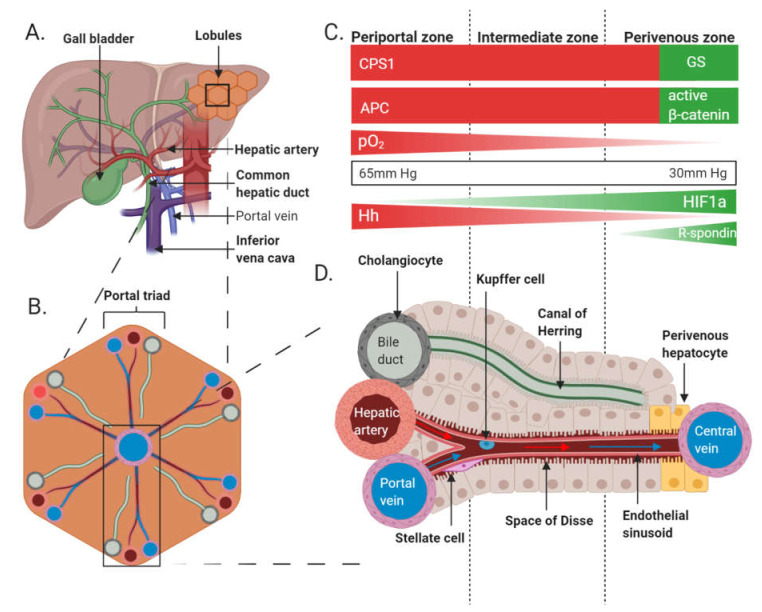
Structural and functional zonation in the adult liver. (**A**) The liver receives oxygenated blood from the heart via the hepatic artery (red; 25%) and deoxygenated blood via the portal vein (blue; 75%). Blood drains via the inferior vena cava (purple). Bile drains from the liver and gallbladder via the common hepatic duct (green). (**B**) Liver lobules are roughly hexagonal. The portal triads comprise the portal vein, hepatic artery and bile duct and are found at each vertex surrounding a central vein. (**C**) The acinus is divided into 3 zones: periportal, intermediate and perivenous. APC and carbamoyl phosphate synthase 1 (CPS1) are expressed throughout the periportal and intermediate zones in a mutually exclusive relationship with glutamine synthetase (GS) which is expressed only in the last 1–2 layers of hepatocytes in the perivenous zone as result of the transcriptional activity of nuclear β-catenin. Wnt/β-catenin signalling is, therefore, largely restricted to the perivenous zone. Oxygen (O_2_) concentration (pO_2_) decreases as blood moves through the acinus and is lowest in the perivenous zone. HIF activity is highest in the perivenous region. Hh ligands are secreted by periportal endothelial cells. (**D**) The acinus comprises a sinusoid connecting the hepatic artery and portal vein with the central vein. Oxygenated blood from the hepatic artery mixes with deoxygenated blood from the portal vein as it flows towards the central vein. Stellate cells and Kupffer cells are present throughout the endothelial sinusoid. Bile secreted from the apical membrane of hepatocytes drains into the canal of Herring which empties into the bile duct. The basolateral membranes of hepatocytes line the sinusoid and project into the space of Disse where proteins and metabolites are exchanged. Stellate cells are found in the space of Disse, a cavity which separates hepatocytes from the sinusoidal endothelial cells. Kupffer cells, the resident macrophage cell, are present within the endothelial sinusoid.

**Figure 3 genes-11-01163-f003:**
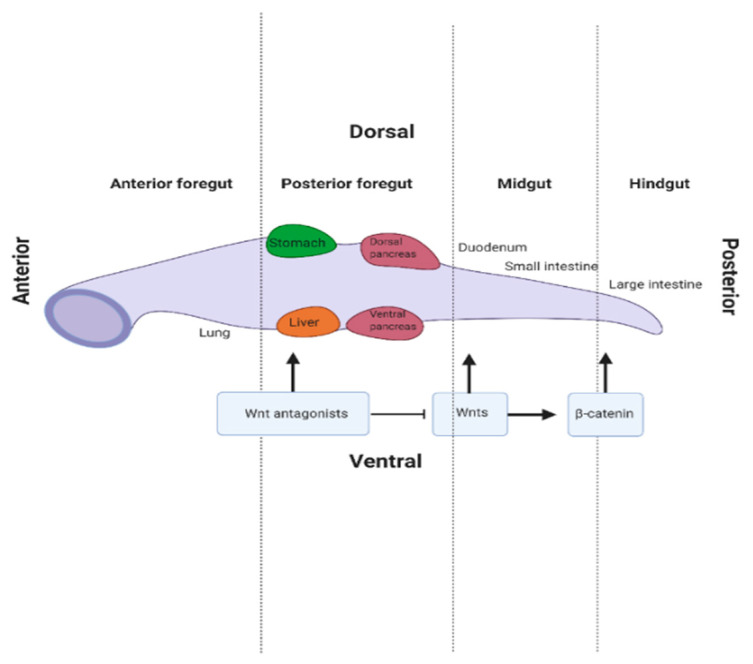
Role of Wnt/β-catenin signalling in gut tube patterning. After gastrulation, the definitive endoderm (DE) forms into the primitive gut tube along the anterior–posterior axis. This gives rise to the foregut, midgut and hindgut domains. Wnt/β-catenin signalling acts by inducing midgut and hindgut development in the posterior axis. Foregut formation occurs when Wnt antagonists secreted from the anterior endoderm suppress Wnt/β-catenin signalling, allowing for subsequent liver development.

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
