# Peer review of "The Canonical Wnt Pathway as a Key Regulator in Liver Development, Differentiation and Homeostatic Renewal"

_genes, 2020, doi:10.3390/genes11101163_

Round 1

Reviewer 1 Report

The canonical Wnt/beta-catenin pathway is one of the principal signaling mechanisms involved in the control of stem cell behavior, differentiation, development and homeostasis in a majority of vertebrate organ systems.  As such, the literature relating its functions in individual organs is forever expanding and worthy of periodic update and review, as intended here.  The present manuscript, with over 200 references, is nothing if not comprehensive in scope.  While this might be seen as a positive feature, it also makes the article undesirably long.  The length is compounded by focusing on two organs, liver and pancreas.  Although both of endodermal origin, the parallels drawn between the two, in terms of Wnt signaling, are modest at best and, while there have been important major updates in the liver story, this is not so evident for pancreas.  The case for combining the two subjects into one review is weak, therefore.  Moreover, the quality of the writing is very variable, perhaps reflecting multiple authors, and much of the pancreas section is badly organized and of poor quality overall.  The opinion of this reviewer is that the readership would be much better served by the article concentrating only on the liver, and trying to provide the big picture of what Wnt signaling does there, emphasizing guiding principles and staying close to direct effects of Wnt signaling rather than secondary effects or links with additional degrees of separation.  Especially useful would be attempts to reconcile conflicting data, such as on the key issue of liver regrowth after injury. 

Although mostly comprehensive in scope, one aspect that seems missing from the manuscript is to describe the spatial and temporal highlights of canonical Wnt signaling as defined by various mouse reporter strains that have bee3n employed over the years (TOPgal, BATgal, Axin2-lacZ).  While these are neither infallible nor comprehensive, they do provide powerful in vivo corroboration of Wnt effects otherwise demonstrated by less physiological experimental systems.

Although the manuscript starts well in terms of clarity and accuracy, those aspects tend to worsen later on.  There are also significant numbers of grammatical errors and typos.  What follows below is a summary list of specific problems listed by line number, mostly minor changes that should have been caught at the proof-reading stage, but also some more substantive comments.  Italics are used instead of quotation marks.

Line number    Change

43                    delete and at end of line

60                    Why Wnt3a?  The text says it is Wnt2 and Wnt9b that are expressed in the perivenous cells

62                    replace induces with permits

80-82               This evidence alone does not implicate Wnt signaling because beta-catenin has an essential role in cell-cell adhesion that is not generally regulated by Wnts.  Crucial roles of Wnts in embryogenesis are evident from many fo the knowck-out phenotypes of individual mouse Wnts, however.

89                    I don’t understand what is meant by overlap in the development of tumors.

101                  apostrophe after others

106                  References missing.

118                  No comma after composite.

122                  b-cells should be beta-cells.  Here and about 20 incidences subsequently...

142                  Poor choice of reference.  Use Clevers & Nusse 2012 or one more recent.

146                  context is sufficient

196                  Reference Figure 2.

203                  Comma after tissue.

204                  Comma after (BMP).

231/2              delete text in parentheses.

233                  has been reported

236                  Add reference after proliferation.

240                  No comma after this

250/1              What is SFRP Dickkopf-related protein 1?  Ref 62 is the wrong reference.

254                  higher when treated with DKK1

258                  delete the

259                  change pathway to signaling.  There are at least 3 distinct pathways of non-canonical Wnt signaling, not just one.

270                  while non-canonical Wnt signaling is

276                  activity appears to promote

277                  Thus, Wnt3a

284                  Insert normally after must.

287                  Insert ultimately before differentiate

294/5              Delete this sentence (redundant).

304                  and generated viable HLCs

318                  Delete for the first time

452                  Change morphogens to agonists

457                  Change adherent protein to cell adhesion protein

467                  potentiating

469                  receptor, thus preventing

470                  delete over

474                  add comma after zonation

480                  adhesion protein

483-490           Too much detail here.

502-504           Missing references here.

508-514           Too confusing.  Omit these 3 sentences.

535                  change effector to target.  (and expression of Axin2 does NOT maintain their proliferative ability)

554                  change effector to target

552-568           This important section needs work.  Currently it fails to explain the difference in conclusions reached by refs 80 and 98.  There are 2-3 Cell Stem Cell papers in Jan 2020 that cover this controversy.  By the end of this paragraph the reader will want to know what the consensus or prevailing opinion is.  (Is there any evidence that the Axin2-lacZ reporter is haploinsufficient?  Or is that just conjecture?)

575                  pathways help

570-657           The long coverage of these secondary effects of Wnt signaling, through other signals, could be greatly reduced or even omitted.

Part 2: Pancreas

I am not going to enumerate individual corrections or suggestions in this section as I believe it should be omitted altogether so the review can focus on liver.  There are numerous grammatical problems and several places where a complete re-write would be needed since the text is badly organized, incomprehensible, or inaccurate.  All of page 18, and the sections about directed differentiation on pages 19-20, are particularly poor.

Author Response

Subject: Revised review publication in MDPI Genes – Special issue Wnt Signalling in Development, Regeneration and cancer

 Manuscript Title: The Canonical Wnt Pathway as a Key Regulator in Liver Development, Differentiation and Homeostatic Renewal.

 Manuscript ID: 916760

Dear Luke,

Thank you for the reviewers’ comments on our manuscript entitled ‘The Canonical Wnt Pathway as a Key Regulator in Liver Development, Differentiation and Homeostatic Renewal’ (revised title) submitted for publication in the Wnt Signalling in Development, Regeneration and Cancer special issue in the journal Genes.

We wish to express our grateful thanks to the reviewers for their constructive and insightful comments. We have endeavoured to address all comments and suggestions and hope that we have been able to satisfy all your concerns. Please find below a list of reviewer comments and our responses. The resubmitted manuscript also contains tracked changes. In light of the fact that the manuscript has undergone major revisions, particularly the omittance of text pertaining to discussion of pancreas, large portions of text have been deleted and some text has been rearranged. We also attach a clear version of the review (with track changes accepted).

We hope that you will now find the revised manuscript suitable for publication. We thank you for your time, patience and consideration.

We look forward to hearing from you.

Yours faithfully,

Sebastian Wild, Aya Elghajiji, Carmen Rodriguez Grimaldos, Stephen Weston, Zoë Burke and David Tosh

Reviewer #1 comments and author responses

 We thank the reviewer for their constructive comments. The comments are copied below together with our responses.

General comments:

The canonical Wnt/beta-catenin pathway is one of the principal signaling mechanisms involved in the control of stem cell behavior, differentiation, development and homeostasis in a majority of vertebrate organ systems.  As such, the literature relating its functions in individual organs is forever expanding and worthy of periodic update and review, as intended here.  The present manuscript, with over 200 references, is nothing if not comprehensive in scope.  While this might be seen as a positive feature, it also makes the article undesirably long.  The length is compounded by focusing on two organs, liver and pancreas.  Although both of endodermal origin, the parallels drawn between the two, in terms of Wnt signaling, are modest at best and, while there have been important major updates in the liver story, this is not so evident for pancreas.  The case for combining the two subjects into one review is weak, therefore.  Moreover, the quality of the writing is very variable, perhaps reflecting multiple authors, and much of the pancreas section is badly organized and of poor quality overall.  The opinion of this reviewer is that the readership would be much better served by the article concentrating only on the liver, and trying to provide the big picture of what Wnt signaling does there, emphasizing guiding principles and staying close to direct effects of Wnt signaling rather than secondary effects or links with additional degrees of separation.  Especially useful would be attempts to reconcile conflicting data, such as on the key issue of liver regrowth after injury.

 Although mostly comprehensive in scope, one aspect that seems missing from the manuscript is to describe the spatial and temporal highlights of canonical Wnt signaling as defined by various mouse reporter strains that have been employed over the years (TOPgal, BATgal, Axin2-lacZ).  While these are neither infallible nor comprehensive, they do provide powerful in vivo corroboration of Wnt effects otherwise demonstrated by less physiological experimental systems.

 Although the manuscript starts well in terms of clarity and accuracy, those aspects tend to worsen later on.  There are also significant numbers of grammatical errors and typos.  What follows below is a summary list of specific problems listed by line number, mostly minor changes that should have been caught at the proof-reading stage, but also some more substantive comments.  Italics are used instead of quotation marks.

Response: We accept the criticism of the length of the paper and have removed discussion of the pancreas from the manuscript as well as reducing in size the section on cross-talk. The review now focuses on the role of canonical Wnt signalling in liver development and homeostasis. We have sought to place a greater emphasis on recent important developments pertaining to homeostatic renewal and regeneration. We also have included additional references to studies utilising transgenic mice, particularly relating to loss of function studies though have stopped short of expanding the review to explicitly discuss the contribution of reporter strains owing to length considerations. We do discuss several recent papers which make use of other inducible knockout and reporter strains. With respect to grammatical and typographical errors we thank the Reviewer for highlighting these and these have now been corrected. All comments relating to missing references, consistency and language have been amended in line with reviewer suggestions.

Reviewer #1 additional comments and author responses

Line number    Change

80-82               This evidence alone does not implicate Wnt signaling because beta-catenin has an essential role in cell-cell adhesion that is not generally regulated by Wnts.  Crucial roles of Wnts in embryogenesis are evident from many of the knock-out phenotypes of individual mouse Wnts, however.

Response: We have updated this reference and included additional references relating to the contribution of knock out studies to our understanding of the canonical Wnt pathway

142                 

146                  context is sufficient

Response: Replaced with “The subject of this review will focus on contextualising recent experimental findings within the literature, exploring the role of Wnt/β-catenin signalling in orchestrating hepatic development and homeostasis.”

483-490           Too much detail here.

Response: Unnecessary detail removed.

508-514           Too confusing.  Omit these 3 sentences.

Response: Sentences have been omitted.

535                  change effector to target.  (and expression of Axin2 does NOT maintain their proliferative ability)

Response: This has been changed.

552-568           This important section needs work.  Currently it fails to explain the difference in conclusions reached by refs 80 and 98.  There are 2-3 Cell Stem Cell papers in Jan 2020 that cover this controversy.  By the end of this paragraph the reader will want to know what the consensus or prevailing opinion is.  (Is there any evidence that the Axin2-lacZ reporter is haploinsufficient?  Or is that just conjecture?)

Response: This section has been revised to include the suggested references and clarify the differences in conclusions. We have sought to make clear that the suggestion of haploinsufficiency, though not empirically demonstrated, has a scientific grounding and may explain the disparity in findings.

570-657           The long coverage of these secondary effects of Wnt signalling, through other signals, could be greatly reduced or even omitted.

Response: These sections have been consolidated and reduced under one heading.

Part 2: Pancreas

I am not going to enumerate individual corrections or suggestions in this section as I believe it should be omitted altogether so the review can focus on liver.  There are numerous grammatical problems and several places where a complete re-write would be needed since the text is badly organized, incomprehensible, or inaccurate.  All of page 18, and the sections about directed differentiation on pages 19-20, are particularly poor.

Response: The pancreas section has been omitted.

Reviewer #1 comments and author responses

 We thank the reviewer for their kind and constructive comments. The comments are copied below together with our responses. Please note we have removed discussion of the pancreas as per Reviewer 1’s comments.

 General comments:

The paper is an extensive review of influence of canonical Wnt pathway on development of Liver and pancreas. It is concisely written, and despite its length it reads with ease. I did not find any major issues so only some minor comments.

Reviewer #2 minor comments and author responses

Authors should consider whether the abbreviation cWnt should not be replaced by Wnt/β-catenin pathway throughout the article as this is broadly used by the community, it is a first time i have seen the use of the abbreviation cWnt.”

Response: cWnt has been replaced with Wnt/β-catenin throughout the document.

Line 74: Translocation of SMO is not to the plasma membrane itself but rather to the membrane of primary cilium, organelle upon which the HH signaling is crucially dependent on. Authors should add at least a sentence about HH signaling being dependent on primary cilia, as the information presented here is incomplete”

Response: We thank the Reviewer for pointing this out and have amended accordingly.

line 82: authors should mention that the b-catenin is also a tight junction protein and the lethatlity of the whole embryo knock-out is partially due to its loss from this structure. there is a b-catenin mutant available that separates its roles in Wnt/b catenin signaling from roles in cytoskeleton. reviewed in Valenta2012Embo Journal: The many faces and functions of β-catenin. authors should consider adding a mention of these two functions of this protein and to elaborate on this here”

Response: We thank the Reviewer for pointing this out and have amended accordingly.

b-cells versus β-cells throughout the manuscript, but noticed on lines 122, 144, 784-810 many times, 856....

Response: Amended. The review now only focuses on canonical Wnt signalling in Liver development and homeostasis as per Reviewer 1 request.

line 709: abbreviation Frz is not defined (for frizzled), consider using more common Fz

Response: Amended to FZD throughout

Reviewer 2 Report

Summary:

The paper is an extensive review of influence of canonical Wnt pathway on development of Liver and pancreas. It is concisely written, and despite its length it reads with ease. I did not find any major issues so only some minor comments:

Points to consider:

Authors should consider whether the abbreviation cWnt should not be replaced by Wnt/β-catenin pathway throughout the article as this is broadly used by the community, it is a first time i have seen the use of the abbreviation cWnt.

Line 74: Translocation of SMO is not to the plasma membrane itself but rather to the membrane of primary cilium, organelle upon which the HH signaling is crucially dependent on. Authors should add at least a sentence about HH signaling being dependent on primary cilia, as the information presented here is incomplete

line 82: authors should mention that the b-catenin is also a tight junction protein and the lethatlity of the whole embryo knock-out is partially due to its loss from this structure. there is a b-catenin mutant available that separates its roles in Wnt/b catenin signaling from roles in cytoskeleton. reviewed in Valenta2012Embo Journal: The many faces and functions of β-catenin. authors should consider adding a mention of these two functions of this protein and to elaborate on this here

Minor:

b-cells versus β-cells throughout the manuscript, but noticed on lines 122, 144, 784-810 many times, 856....

line 709: abbreviation Frz is not defined (for frizzled), consider using more common Fz

Line 783: heading not in italic

Author Response

Subject: Revised review publication in MDPI Genes – Special issue Wnt Signalling in Development, Regeneration and cancer

 Manuscript Title: The Canonical Wnt Pathway as a Key Regulator in Liver Development, Differentiation and Homeostatic Renewal.

 Manuscript ID: 916760

Dear Luke,

Thank you for the reviewers’ comments on our manuscript entitled ‘The Canonical Wnt Pathway as a Key Regulator in Liver Development, Differentiation and Homeostatic Renewal’ (revised title) submitted for publication in the Wnt Signalling in Development, Regeneration and Cancer special issue in the journal Genes.

We wish to express our grateful thanks to the reviewers for their constructive and insightful comments. We have endeavoured to address all comments and suggestions and hope that we have been able to satisfy all your concerns. Please find below a list of reviewer comments and our responses. The resubmitted manuscript also contains tracked changes. In light of the fact that the manuscript has undergone major revisions, particularly the omittance of text pertaining to discussion of pancreas, large portions of text have been deleted and some text has been rearranged. We also attach a clear version of the review (with track changes accepted).

We hope that you will now find the revised manuscript suitable for publication. We thank you for your time, patience and consideration.

We look forward to hearing from you.

Yours faithfully,

Sebstian Wild, Aya Elghajiji, Carmen Rodriguez Grimaldos, Stephen Weston, Zoë Burke and David Tosh

Reviewer #1 comments and author responses

 We thank the reviewer for their constructive comments. The comments are copied below together with our responses.

General comments:

The canonical Wnt/beta-catenin pathway is one of the principal signaling mechanisms involved in the control of stem cell behavior, differentiation, development and homeostasis in a majority of vertebrate organ systems.  As such, the literature relating its functions in individual organs is forever expanding and worthy of periodic update and review, as intended here.  The present manuscript, with over 200 references, is nothing if not comprehensive in scope.  While this might be seen as a positive feature, it also makes the article undesirably long.  The length is compounded by focusing on two organs, liver and pancreas.  Although both of endodermal origin, the parallels drawn between the two, in terms of Wnt signaling, are modest at best and, while there have been important major updates in the liver story, this is not so evident for pancreas.  The case for combining the two subjects into one review is weak, therefore.  Moreover, the quality of the writing is very variable, perhaps reflecting multiple authors, and much of the pancreas section is badly organized and of poor quality overall.  The opinion of this reviewer is that the readership would be much better served by the article concentrating only on the liver, and trying to provide the big picture of what Wnt signaling does there, emphasizing guiding principles and staying close to direct effects of Wnt signaling rather than secondary effects or links with additional degrees of separation.  Especially useful would be attempts to reconcile conflicting data, such as on the key issue of liver regrowth after injury.

 Although mostly comprehensive in scope, one aspect that seems missing from the manuscript is to describe the spatial and temporal highlights of canonical Wnt signaling as defined by various mouse reporter strains that have been employed over the years (TOPgal, BATgal, Axin2-lacZ).  While these are neither infallible nor comprehensive, they do provide powerful in vivo corroboration of Wnt effects otherwise demonstrated by less physiological experimental systems.

 Although the manuscript starts well in terms of clarity and accuracy, those aspects tend to worsen later on.  There are also significant numbers of grammatical errors and typos.  What follows below is a summary list of specific problems listed by line number, mostly minor changes that should have been caught at the proof-reading stage, but also some more substantive comments.  Italics are used instead of quotation marks.

Response: We accept the criticism of the length of the paper and have removed discussion of the pancreas from the manuscript as well as reducing in size the section on cross-talk. The review now focuses on the role of canonical Wnt signalling in liver development and homeostasis. We have sought to place a greater emphasis on recent important developments pertaining to homeostatic renewal and regeneration. We also have included additional references to studies utilising transgenic mice, particularly relating to loss of function studies though have stopped short of expanding the review to explicitly discuss the contribution of reporter strains owing to length considerations. We do discuss several recent papers which make use of other inducible knockout and reporter strains. With respect to grammatical and typographical errors we thank the Reviewer for highlighting these and these have now been corrected. All comments relating to missing references, consistency and language have been amended in line with reviewer suggestions.

Reviewer #1 additional comments and author responses

Line number    Change

80-82               This evidence alone does not implicate Wnt signaling because beta-catenin has an essential role in cell-cell adhesion that is not generally regulated by Wnts.  Crucial roles of Wnts in embryogenesis are evident from many of the knock-out phenotypes of individual mouse Wnts, however.

Response: We have updated this reference and included additional references relating to the contribution of knock out studies to our understanding of the canonical Wnt pathway

142                 

146                  context is sufficient

Response: Replaced with “The subject of this review will focus on contextualising recent experimental findings within the literature, exploring the role of Wnt/β-catenin signalling in orchestrating hepatic development and homeostasis.”

483-490           Too much detail here.

Response: Unnecessary detail removed.

508-514           Too confusing.  Omit these 3 sentences.

Response: Sentences have been omitted.

535                  change effector to target.  (and expression of Axin2 does NOT maintain their proliferative ability)

Response: This has been changed.

552-568           This important section needs work.  Currently it fails to explain the difference in conclusions reached by refs 80 and 98.  There are 2-3 Cell Stem Cell papers in Jan 2020 that cover this controversy.  By the end of this paragraph the reader will want to know what the consensus or prevailing opinion is.  (Is there any evidence that the Axin2-lacZ reporter is haploinsufficient?  Or is that just conjecture?)

Response: This section has been revised to include the suggested references and clarify the differences in conclusions. We have sought to make clear that the suggestion of haploinsufficiency, though not empirically demonstrated, has a scientific grounding and may explain the disparity in findings.

570-657           The long coverage of these secondary effects of Wnt signalling, through other signals, could be greatly reduced or even omitted.

Response: These sections have been consolidated and reduced under one heading.

Part 2: Pancreas

I am not going to enumerate individual corrections or suggestions in this section as I believe it should be omitted altogether so the review can focus on liver.  There are numerous grammatical problems and several places where a complete re-write would be needed since the text is badly organized, incomprehensible, or inaccurate.  All of page 18, and the sections about directed differentiation on pages 19-20, are particularly poor.

Response: The pancreas section has been omitted.

Reviewer #1 comments and author responses

 We thank the reviewer for their kind and constructive comments. The comments are copied below together with our responses. Please note we have removed discussion of the pancreas as per Reviewer 1’s comments.

 General comments:

The paper is an extensive review of influence of canonical Wnt pathway on development of Liver and pancreas. It is concisely written, and despite its length it reads with ease. I did not find any major issues so only some minor comments.

Reviewer #2 minor comments and author responses

Authors should consider whether the abbreviation cWnt should not be replaced by Wnt/β-catenin pathway throughout the article as this is broadly used by the community, it is a first time i have seen the use of the abbreviation cWnt.”

Response: cWnt has been replaced with Wnt/β-catenin throughout the document.

Line 74: Translocation of SMO is not to the plasma membrane itself but rather to the membrane of primary cilium, organelle upon which the HH signaling is crucially dependent on. Authors should add at least a sentence about HH signaling being dependent on primary cilia, as the information presented here is incomplete”

Response: We thank the Reviewer for pointing this out and have amended accordingly.

line 82: authors should mention that the b-catenin is also a tight junction protein and the lethatlity of the whole embryo knock-out is partially due to its loss from this structure. there is a b-catenin mutant available that separates its roles in Wnt/b catenin signaling from roles in cytoskeleton. reviewed in Valenta2012Embo Journal: The many faces and functions of β-catenin. authors should consider adding a mention of these two functions of this protein and to elaborate on this here”

Response: We thank the Reviewer for pointing this out and have amended accordingly.

b-cells versus β-cells throughout the manuscript, but noticed on lines 122, 144, 784-810 many times, 856....

Response: Amended. The review now only focuses on canonical Wnt signalling in Liver development and homeostasis as per Reviewer 1 request.

line 709: abbreviation Frz is not defined (for frizzled), consider using more common Fz

Response: Amended to FZD throughout

Round 2

Reviewer 1 Report

The authors have responded comprehensively to the reviewers' recommendations and the revised version is hugely improved and clearer to understand.  A useful review of the subject.

There remain some minor grammatical issues, which are itemized below by line number.

23  organ, not organs

44  pathway, not pathways

49  insert 'is' before 'also'

60  Add 'in the liver' to title of Figure.

70  ...'where they dimerise with'...

71  form, not from

177  comma after 'begins'

215  insert 'the' after 'activate'

233  consist and comprise should both be singular

316  better to change 'arises' to 'arose'

321  RNA, however, was not affected.

370  add comma after 'zonation'

385  add comma after RSPO3

412  insert comma after 'tracing'

424  supports should be plural

440  proffer a potential rebuttal  (don't take sides too much!)

474  insert comma after 'liver'

480  insert comma after 'mice'

526  delete 'inhibitors such as'  (they are not Wnt inhibitors).

527  add reference at end of line

532  express, not expressing

533  express, not expressed

534  comaas after 'signalling' and 'compartment'

541  insert 'a' before 'model'

556  insert comma after 'profiles'